# DREAM-R: Multimodal Speculative Reasoning with RL-Based Refined Drafting, Precise Verification, and Fully Parallel Execution

Yunhai Hu [1]  Zining Liu [2]  Xiangyang Yin [1]  Tianhua Xia [1]
Bo Bao [3]  Eric Sather [3]  Vithursan Thangarasa [3]  Sai Qian Zhang [1]

## Abstract

Speculative reasoning has recently been proposed as a means to accelerate reasoning-intensive generation in large multimodal models, but its effectiveness is often constrained by misalignment between speculative drafts and target-verified reasoning. In this work, we introduce *DREAM-R*, a framework that substantially improves the performance of speculative reasoning. At its core, DREAM-R employs *Speculative Alignment Policy Optimization* (SAPO), a reinforcement-learning objective that trains draft models to generate reasoning steps that are both faithful to target trajectories and concise. We further propose a *Contrastive Probability Normalization* (CPN) that uses a ratio-based criterion to provide stable and interpretable acceptance of speculative steps only when positive evidence clearly dominates, thereby preventing error propagation. Building on these components, we develop a *Fully Parallel Speculative Reasoning* (FPSR) framework that parallelizes draft generation, target-side reasoning, and verification across multi-step reasoning, enabling early stopping and clean fallback. Experiments on reasoning-heavy benchmarks demonstrate up to $2.48\times$ speedup while preserving target-model accuracy, yielding substantial efficiency gains without compromising reasoning quality.

## 1. Introduction

Large models (LMs) (Grattafiori et al., 2024; Brown et al., 2020; Ouyang et al., 2022; Achiam et al., 2023) have become the backbone of modern artificial intelligence, driving major advances across language understanding, vision, and multimodal perception by leveraging massive model capacity and data scale. While these models excel at representation learning and pattern recognition, many real-world tasks require more than direct prediction, demanding structured, multi-step inference and long-horizon decision-making. This need has given rise to large reasoning models (LRMs) (Bai et al., 2025; Hong et al., 2025; Wang et al., 2025a), which extend large models with explicit reasoning capabilities that can decompose complex problems, integrate heterogeneous evidence, and maintain consistency across extended inference chains. LRMs are particularly important in settings such as multimodal understanding, scientific problem solving, and embodied intelligence, where accurate outcomes depend on coherent reasoning rather than surface-level correlations. However, the increased computational cost of reasoning-intensive generation presents significant efficiency challenges, motivating research into methods that can accelerate reasoning while preserving accuracy.

To address this challenge, recent advances in speculative decoding (Li et al., 2024a;b; 2025b; Cai et al., 2024; Ankner et al., 2024; Xia et al., 2023; Zhang et al., 2023; Miao et al., 2023; Chen et al., 2024; Sun et al., 2024; Hu et al., 2025a;b) have shown promise in accelerating inference by parallelizing generation and verification using lightweight draft models. Building on this idea, speculative reasoning (Pan et al., 2025; Fu et al., 2025) extends speculation beyond token-level decoding to reasoning-step granularity, enabling more effective acceleration for LRMs by exploiting parallelism while preserving correctness. These developments point toward a promising direction for scaling reasoning performance without proportionally increasing inference cost.

Most existing speculative decoding and speculative reasoning methods are developed and evaluated primarily for large language models (LLMs) and LRMs, with limited investigation into multimodal LLMs (MLLMs), especially multimodal large reasoning models (MLRMs). In practice, we observe that directly applying these techniques to MLRMs often yields suboptimal performance, as the reasoning characteristics and error modes in multimodal settings differ substantially from those of text-only models. Through an empirical analysis of multimodal reasoning behavior, we observe that design choices effective for LLMs do not trans-

[1] New York University [2] University of Pennsylvania [3] Cerebras Systems. Correspondence to: Yunhai Hu <yh5961@nyu.edu>.

fer reliably to MLRMs. Motivated by these observations, we introduce *DREAM-R*, a multimodal speculative reasoning framework that integrates RL-based refined drafting, precise threshold-based verification, and fully parallel execution to effectively accelerate multimodal reasoning while preserving accuracy. In particular, our contribution can be summarized as follows:

- DREAM-R incorporates a *Speculative Alignment Policy Optimization* (SAPO), a reinforcement-learning objective that trains draft models to produce reasoning steps that are both faithful to target-model trajectories and concise.

- Furthermore, we introduce a *Contrastive Probability Normalization* (CPN) that employs a contrastive probability normalization to ensure stable and interpretable acceptance of speculative steps only when positive evidence clearly outweighs negative evidence, effectively preventing error propagation.

- Finally, we present a *Fully Parallel Speculative Reasoning* (FPSR) framework that parallelizes draft generation, target-side reasoning, and verification across multi-step reasoning, enabling early stopping and clean fallback.

- Experiments on reasoning-heavy benchmarks demonstrate up to $2.48\times$ speedup while maintaining target-model accuracy, delivering substantial efficiency gains without compromising reasoning quality.

## 2. Related Works

**Large Reasoning Models** Large reasoning models (LRMs) are distinguished by their ability to generate extended chains of thought (CoT) (Wei et al., 2022), enabling explicit intermediate reasoning steps prior to producing a final answer. In language-only settings, this inference paradigm has yielded substantial gains on complex tasks such as mathematical reasoning (Cobbe et al., 2021) and code generation (Chen, 2021). For vision-language reasoning models (VLRMs), CoT reasoning naturally extends to multimodal scenarios that require integrating visual evidence with textual inference. Recent VLRMs (Bai et al., 2025; Hong et al., 2025; Wang et al., 2025a) demonstrate strong performance on vision-dependent reasoning benchmarks, highlighting the effectiveness of multimodal CoT reasoning.

However, VLRMs typically incur high decoding latency, which is largely driven by the generation of intermediate reasoning steps. Analyzing Qwen3-VL-4B, Qwen3-VL-32B, and Qwen3-VL-235B on 50 samples from the MathVerse dataset, we find that the models generate an average of 2,330

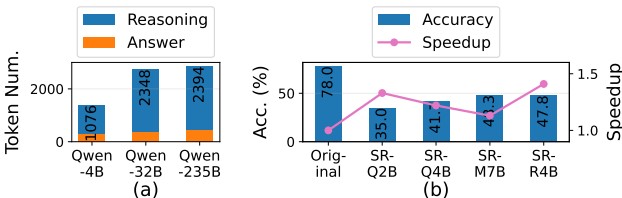

*Figure 1.* (a) Numbers of reasoning and answer tokens. Qwen-4B, Qwen-32B, and Qwen-235B refer to Qwen3-VL-4B, Qwen3-VL-32B, and Qwen3-VL-235B-A22B. (b) Accuracy and speedup of Qwen3-VL-32B under different decoding methods. Original denotes standard decoding. SR-Q2B, SR-Q4B, SR-M7B, and SR-R4B denote SpecReason (Pan et al., 2025) using Qwen3-VL-2B, Qwen3-VL-4B, MiMo-VL-7B-RL, and Qwen3-VL-R1-VL-4B as draft models, respectively. Speedup is normalized to Original.

tokens per sample, with 1,939 tokens corresponding to intermediate reasoning, accounting for approximately 83% of the total output. In contrast, answer tokens make up only 17% of the output, as illustrated in Figure 1(a). These results highlight that the reasoning phase dominates decoding cost and constitutes the primary bottleneck for end-to-end inference latency in VLRMs.

**Speculative Decoding** Speculative decoding accelerates autoregressive inference by pairing a lightweight draft model with a larger target model (Stern et al., 2018). After the target model completes prefilling, the draft model proposes multiple speculative tokens, which are verified in parallel by the target model using causal attention. If all speculative tokens are accepted, decoding advances without additional target computation; otherwise, decoding resumes from the first rejected token.

Prior work has explored speculative decoding along multiple axes (Xia et al., 2023; Hu et al., 2025a). Algorithmically, some methods replace the draft model with retrieval-based (Yang et al., 2023; He et al., 2023) or n-gram-based generators (Ou et al., 2024; Stewart et al., 2024). Architecturally, self-speculative approaches reuse shallow target layers or apply layer skipping to generate drafts (Zhang et al., 2023; Xia et al., 2024; Elhoushi et al., 2024; Liu et al., 2024b), while others introduce lightweight trainable draft heads that share embeddings or output layers with the target model (Li et al., 2024a; Gao et al., 2024b). Verification strategies range from sequential acceptance to tree-based parallel verification (Miao et al., 2023; Chen et al., 2024), and recent systems work explores asynchronous or pipelined execution to improve hardware utilization (Chen et al., 2023; McDanel et al., 2025).

**Multimodal Acceleration** Extending speculative decoding to multimodal models introduces additional challenges, as draft generation must account for both linguistic and visual evidence. Several recent works adapt speculative

decoding to multimodal settings by modifying draft model architectures or training strategies. VADUSA (Li et al., 2025a) applies speculative decoding with tolerance mechanisms in text-to-speech systems. For vision-language models, Gagrani et al. (2024) pair a language-only draft model with a VLM target, while IbED (Lee et al., 2025) improves efficiency through batch-level ensembling without additional parameters. Other approaches focus on strengthening multimodal draft models: DREAM (Hu et al., 2025b) incorporates cross-attention and adaptive feature selection, ViSpec (Kang et al., 2025) compresses visual tokens via lightweight vision adapters, and MASSV (Ganesan et al., 2025) adapts small language models into multimodal drafters through architectural changes and self-distillation.

**Speculative Reasoning** Large reasoning models, including both language-only and multimodal variants, achieve strong performance on complex tasks but incur high inference latency due to extended CoT generation. To alleviate this overhead, recent work proposes speculative reasoning (Pan et al., 2025), where a lightweight draft model generates candidate reasoning steps that are verified by a larger target model. The target model assigns a scalar score to each step to determine acceptance. The key insight is that many intermediate reasoning steps do not require the full capacity of the target model and can be accurately produced by a smaller model, while preserving semantic correctness even when token-level outputs differ.

Building on this line of work, Lookahead Reasoning (Fu et al., 2025) proposes an asynchronous architecture that overlaps the drafting and verification stages, allowing the target model to verify earlier steps while new reasoning steps are generated in parallel. However, these approaches are developed and evaluated primarily for language-only models, and our experiments indicate that directly extending them to vision-language models provides limited benefits. To better understand this limitation, we conduct a simple diagnostic study using 1,000 samples from the MathVista dataset, focusing on the behavior of speculative reasoning when there is a large capability gap between the draft and target models. We evaluate the SpecReason (Pan et al., 2025) method using Qwen3-VL-32B as the target model and Qwen3-VL-2B, Qwen3-VL-4B, MiMo-VL-7B-RL, and R-4B as draft models, respectively. We compare the accuracy and speedup of SpecReason against the standard decoding baseline. As shown in Figure 1 (b), SpecReason's accuracy drops from 78% to 43.2%. One contributing factor is perceptual error in vision-language reasoning, where failures stem from incorrect or incomplete visual grounding rather than flawed logic. For example, in the blue tape positioning case, the visual cue exists in the image but is mis-recognized by the draft model, leading to a coherent yet visually unsupported reasoning trace that complicates verification. This highlights that vi-

sual reasoning fundamentally requires grounding in images, which standard language-model drafts tend to ignore.

We find that when the target model is a vision–language model, its judgments of reasoning correctness follow different logic and distributions from those of standard language models. As a result, directly using the log-probabilities of its step-level scores as a reliability signal becomes inaccurate and unstable.

**Reinforcement-Learning Based Large Model Training**
Reinforcement learning (RL) has emerged as a central component of the post-training pipeline for foundation models, including both LLMs and VLMs (Grattafiori et al., 2024; Bai et al., 2025; Liu et al., 2024a; Guo et al., 2025; Comanici et al., 2025). Through alignment, recent models such as DeepSeek-R1 (Liu et al., 2024a; Guo et al., 2025) have demonstrated substantial performance improvements across a broad range of downstream tasks, particularly those involving complex multi-step reasoning. Existing RL-based alignment approaches can be broadly categorized based on the source of their reward signals into two paradigms: reinforcement learning from human feedback (RLHF) (Ziegler et al., 2019; Stiennon et al., 2020; Ouyang et al., 2022; Bai et al., 2022; Rafailov et al., 2023; Sun et al., 2023; Ethayarajh et al., 2024) and reinforcement learning with verifiable rewards (RLVR) (Gao et al., 2024a; Zeng et al., 2025; Guan et al., 2025; Wang et al., 2025b; Cui et al., 2025; Yu et al., 2025). RLHF leverages human preference annotations to guide alignment, whereas RLVR derives reward signals from automated or programmatic verification mechanisms.

Despite its effectiveness, RLHF depends on human preference annotations and primarily captures relative preferences rather than objective correctness. Extensions such as reinforcement learning from AI feedback (RLAIF) (Bai et al., 2022) help reduce annotation costs, but preference-based signals remain limited in their ability to reliably evaluate correctness for reasoning-intensive tasks, including mathematical problem solving and code generation. In contrast, RLVR relies on task-specific verifiers (Cheng et al., 2024; Kydlíček, 2025) to assess correctness during training, reducing dependence on human feedback and enabling more reliable correctness evaluation.

Following the RLVR paradigm, our work investigates verifier-guided reward design for speculative reasoning, with the goal of improving the quality of draft-generated reasoning steps and, in turn, enhancing both accuracy and latency in speculative reasoning pipelines, as described in Section 3.

## 3. Method

In this section, we first present the *Contrastive Probability Normalization* (CPN) for validating drafted reasoning,

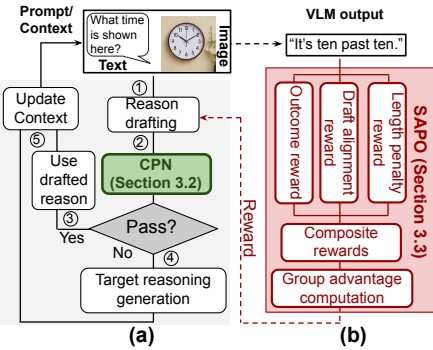

**(a)**          **(b)**

*Figure 2.* (a) The DREAM-R decoding step. All the step numbers are highlighted in circles. (b) A highlight of SAPO training.

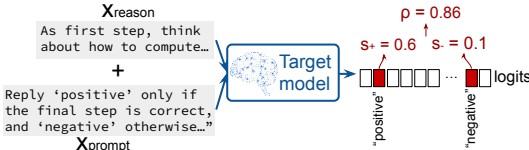

*Figure 3.* An example on CPN.

followed by *Speculative Alignment Policy Optimization* (SAPO), an RL-based alignment method that trains the draft model to produce reasoning steps consistent with multimodal evidence. Finally, we introduce *Fully Parallel Speculative Reasoning* (FPSR), which concurrently performs drafting, target generation, and verification to maximize hardware utilization and reduce wall-clock latency.

### 3.1. Overview of DREAM-R

Figure 2 illustrates the overall architecture of DREAM-R. As shown in Figure 2(a), the draft model $M_{\text{draft}}$ first processes the input sample and generates a CoT reasoning trace (Step 1). The target model $M_{\text{target}}$ then performs CPN to assess the quality of this reasoning (Step 2). If the reasoning produced by $M_{\text{draft}}$ passes the verification check (Step 3), it is fed back to $M_{\text{draft}}$ to update the CoT context for the next execution round (Step 5). Otherwise, if the reasoning fails the CPN check (Step 4), the target model $M_{\text{target}}$ directly generates the CoT reasoning from the input.

### 3.2. Contrastive Probability Normalization

As illustrated in Figure 2(a), CPN is used to assess the quality of the reasoning $x_{\text{reason}}$ generated by $M_{\text{draft}}$. DREAM-R adopts a verification scheme in which $x_{\text{reason}}$, together with a predefined prompt template $x_{\text{prompt}}$, is sent to the target model $M_{\text{target}}$. The prompt $x_{\text{prompt}}$ instructs $M_{\text{target}}$ to evaluate the quality of $x_{\text{reason}}$ by selecting one of two keywords: "positive" or "negative". As shown in Figure 3, let $s_+$ and $s_-$ denote the predicted probabilities of the keywords "positive" and "negative", respectively. The verification decision is then made by computing the ratio $\rho = \frac{s_+}{s_+ + s_-}$.

This ratio $\rho$ captures the relative dominance between the two signals. When the verifier assigns comparable probabilities to both keywords (e.g., a near 1:1 balance), $\rho$ approaches 0.5, indicating insufficient confidence in a positive judgment. Conversely, larger values of $\rho$ reflect stronger confidence in the positive assessment. Empirically, we adopt a conser-

vative policy: a reasoning step $x_{\text{reason}}$ proposed by $M_{\text{draft}}$ is accepted only if $\rho$ exceeds a predefined threshold $\alpha = 0.7$; otherwise, the reasoning is rejected.

Compared with utility score–based methods such as those proposed in (Pan et al., 2025), which require $M_{\text{target}}$ to assign a discrete score (e.g., from 0 to 9) to the proposed reasoning and accept it if the score exceeds a predefined threshold, CPN provides a more principled and fine-grained assessment. Instead of relying on an arbitrary scalar rating, CPN directly captures the confidence of $M_{\text{target}}$'s judgment by examining the predicted probabilities of the verification keywords. This probabilistic formulation enables a smoother and more interpretable acceptance criterion, reduces sensitivity to prompt-specific scoring biases, and avoids ambiguity introduced by coarse-grained discrete scores. As a result, CPN yields more stable and reliable verification decisions across different reasoning steps and inputs, as demonstrated in Section 4.

### 3.3. Speculative Alignment Policy Optimization

To improve the quality of reasoning produced by $M_{\text{draft}}$, we introduce Speculative Alignment Policy Optimization (SAPO), a policy optimization framework designed for synchronous speculative reasoning. SAPO extends group-based policy optimization by incorporating alignment-aware reward signals that explicitly measure the consistency between the draft model's proposed reasoning steps and the target model's verified reasoning trajectory.

As described in Section 3.1, during LRM decoding, $M_{\text{draft}}$ iteratively proposes candidate reasoning steps, which are verified by $M_{\text{target}}$ via CPN. This process yields a complete reasoning trace and a final prediction for each input. SAPO assigns a composite verification reward to the resulting reasoning trajectory $x_{\text{reason}}$, consisting of an outcome reward $R_{\text{outcome}}$ that reflects final-answer correctness, a draft alignment reward $R_{\text{draft}}$ that measures the proportion of draft-generated steps accepted by the target model, and a length penalty reward term $R_{\text{length}}$ that penalizes unnecessarily long reasoning traces.

$$R_{\text{outcome}} = \begin{cases} 1, & \text{if } x_{\text{reason}} \text{ yields a correct answer,} \\ 0, & \text{otherwise,} \end{cases} \quad (1)$$

while $R_{\text{draft}}$ is defined as $R_{\text{draft}} = \frac{N_{\text{accepted}}}{N_{\text{draft}}}$, where $N_{\text{accepted}}$ and $N_{\text{draft}}$ denote the number of rounds in which the draft

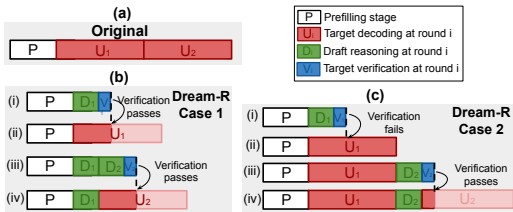

*Figure 4.* Timeline comparison of (a) Standard Autoregressive Reasoning, and (b) Fully Parallel Speculative Reasoning (FPSR). FPSR pipelines the drafting (Green), verification (Blue), and target generation (Red) stages, ensuring the GPU is never idle. Unlike lookahead methods, we allow 'rollback' (c) to recover from verification failures without re-computation.

reasoning is accepted or rejected by the target model. Finally, to discourage unnecessarily long CoT generation, SAPO employs a length-dependent penalty:

$$R_{\text{length}} = \begin{cases} 0, & L \leq C, \\ \min\big(1,\ k \cdot (L - C)\big), & L > C. \end{cases} \quad (2)$$

where $L$ denotes the token length of the reasoning and $C$ is a predefined threshold. Equation 2 imposes a penalty proportional to the excess length beyond $C$. The overall reward is computed as

$$R = w_1 \cdot R_{\text{outcome}} + w_2 \cdot R_{\text{draft}} - w_3 \cdot R_{\text{length}}, \quad (3)$$

where $w_1$, $w_2$, and $w_3$ determine the relative contribution of each component. After computing the rewards, SAPO updates the draft model using a simplified Group Relative Policy Optimization (GRPO) objective (Shao et al., 2024), with the group baseline defined as $b_{\mathcal{G}} = \frac{1}{n} \sum_{i=1}^{N} R_i$, where $N$ is the number of input samples. $R_i$ is the reward corresponding to the $i$-th draft reason. The group-relative advantage for each trajectory is $A_i = R_i - b_{\mathcal{G}}$. To improve stability, advantages $A_i$ are normalized within the group, leading to $\tilde{A}_i = \frac{A_i}{\sigma_{\mathcal{G}} + \delta}$, where $\sigma_{\mathcal{G}}$ denotes the standard deviation of $\{A_i\}$, and $\delta$ is a small constant introduced to improve stability. The policy is updated using a clipped ratio objective:

$$L_i(\theta) = \min\Big(u_i(\theta)\tilde{A}_i,\ \text{clip}(u_i(\theta),\ 1 - \epsilon,\ 1 + \epsilon)\,\tilde{A}_i\Big) \quad (4)$$

where $\theta$ denotes the parameter set. $u_i(\theta) = \frac{\pi_\theta(a_i|x_i)}{\pi_{\theta_{\text{old}}}(a_i|x_i)}$ that follows the definition in (Shao et al., 2024), the overall SAPO process is highlighted in Figure 2 (b).

## 3.4. Fully Parallel Speculative Reasoning

While Sections 3.2 and 3.3 focus on the algorithmic design of DREAM-R to improve the reasoning quality of the LRM, this section introduces an advanced Fully Parallel Speculative Reasoning (FPSR) approach that enables high parallelism across the drafting, verification, and decoding stages to further reduce latency.

The FPSR method is illustrated in Figure 4. For ease of exposition, we assume a total of two rounds of reasoning. Figure 4 (a) depicts the baseline approach, in which the target model $M_{\text{target}}$ sequentially performs the prefilling stage P, generates intermediate reasoning steps $U_1$ and produces the final answer $U_2$, resulting in high end-to-end generation latency.

In contrast, Figure 4 (b) shows the FPSR workflow. At each reasoning round, the draft model $M_{\text{draft}}$ continuously proposes candidate reasoning steps (e.g., $D_1$ in Figure 4 (b)(i)), while the target model $M_{\text{target}}$ concurrently generates its own reasoning ($U_1$ in Figure 4 (b)(ii)). Once $M_{\text{draft}}$ completes a reasoning step, it is immediately verified ($V_1$ in Figure 4 (b)(i)). If the draft reasoning is accepted, $M_{\text{target}}$ early-terminates its ongoing reasoning generation ($U_1$ in Figure 4 (b)(ii)). This process repeats for subsequent reasoning rounds, assuming the draft reasoning continues to pass verification. As a result, FPSR significantly reduces overall latency compared to the baseline scheme shown in Figure 4 (a).

In contrast, when the draft reasoning generated by $M_{\text{draft}}$ fails verification (Figure 4(c)), the subsequent round cannot begin until the current update stage completes ($U_1$ in Figure 4 (c)(ii)). Once $U_1$ finishes, a second reasoning round is initiated ($D_2$ in Figure 4 (c)(iii)) and then verified ($V_2$ in Figure 4 (c)(iii)). Assuming this verification succeeds, the final result is produced in the $D_2$ (Figure 4 (c)(iv)).

Compared to prior parallelization approaches that only overlap target generation (Fu et al., 2025) or speculative reasoning schemes that perform step verification largely sequentially as shown in Figure 4 (a), FPSR maintains continuous parallelism across all three components: the draft model proposing candidate reasoning steps, the target model extending the trusted reasoning prefix, and the verifier determining acceptance or termination. By tightly interleaving proposal, verification, and fallback, FPSR significantly reduces end-to-end wall-clock latency for reasoning-intensive decoding, while fully preserving the target model's answer quality.

## 3.5. DREAM-R System Implementation

We implement a speculative reasoning engine consisting of four tightly coordinated components. The *Drafter* continuously proposes speculative reasoning steps whenever the `can_draft` signal is active, and each proposal is forwarded to the *Verifier* for binary acceptance or rejection. If a step is rejected, the *Corrector* is invoked to produce a revised, non-speculative replacement. A centralized *Manager* orchestrates ordered commitment, rollback, and task cancellation across all components, while enforcing a bounded lookahead window to prevent unbounded speculation.

The control flow proceeds as follows. Drafting runs asynchronously until a rejection occurs. When a speculative step is rejected, all in-flight speculative tasks within the lookahead window are canceled, the system rolls back to the most recent verified prefix, and the corrected step is committed in its place. Drafting then resumes from the updated context. This design enables high-throughput speculative reasoning while strictly preserving logical consistency and stepwise correctness. Detailed prompts are provided in the Appendix B for completeness.

## 4. Experiments

We evaluate our fully parallel speculative reasoning framework on four well-established multimodal reasoning benchmarks, MathVerse (Zhang et al., 2024), MMBench (Liu et al., 2024c), RealWorldQA (xAI, 2024), and MMMU (Yue et al., 2024), using their full evaluation splits. We experiment with multiple draft and target model combinations. The draft model is chosen from Qwen3-VL-2B (Q2B) (Yang et al., 2025a), Qwen3-VL-4B (Q4B), R-4B (R4B) (Yang et al., 2025b), and MiMo-VL-7B-RL (M7B-RL) (Xiaomi et al., 2025). The target model is selected from either Qwen3-VL-32B (Q32B) (Yang et al., 2025a) or Qwen3-VL-235B-A22B (Q235B). All draft and target models are used in their thinking versions.

All draft and target model pairs use identical decoding settings. All evaluations are conducted on NVIDIA L40S GPUs. The target model is deployed on four L40S GPUs using AWQ INT4 quantization, while each draft model runs on two L40S GPUs. We use a fixed speculative lookahead window to 4 and set the acceptance threshold $\alpha$ to 0.7. For each benchmark, we report accuracy, acceptance rate, and speedup relative to target model autoregressive decoding.

The draft model is trained using SAPO on a mixture of multimodal reasoning datasets with step-level annotations, including Geo3K (Lu et al., 2021), OCR-VQA (Mishra et al., 2019), and the ScienceQA (Lu et al., 2022) dev split. Training is conducted on $8\times$NVIDIA H200 GPUs with BF16 precision. We adopt the AdamW optimizer with a peak learning rate of $1 \times 10^{-6}$, a batch size of 64, and train for 15 epochs, using a maximum sequence length of 8196 tokens. The target model is kept frozen throughout training, while the draft model is fully optimized.

To evaluate DREAM-R, we compare it against three baseline methods. The first, *Standard SD*, applies conventional speculative decoding (Leviathan et al., 2023) to LRMs to accelerate generation. We further compare against two recent speculative reasoning approaches, SpecReason (Pan et al., 2025) and LR (Fu et al., 2025). Finally, to isolate the contribution of SAPO, we include a variant of DREAM-R without SAPO, termed *DREAM-R-NS*, which incorporates

*Table 1.* Model performance (%) across four benchmarks.

| Model | MathVerse | MMBench | RealWorldQA | MMMU |
|---|---|---|---|---|
| Q32B | 76.00 | 83.40 | 75.55 | 77.85 |
| Q235B | 85.40 | 83.20 | 75.17 | 71.59 |
| Q2B | 53.94 | 62.90 | 63.29 | 59.27 |
| Q4B | 65.40 | 71.00 | 68.98 | 59.60 |
| M7B-RL | 69.80 | 78.50 | 58.80 | 66.80 |
| R4B | 75.40 | 74.40 | 66.48 | 65.20 |

CPN and FPSR but omits SAPO. Specifically, we adopt three evaluation metrics: model accuracy (Acc.), the acceptance rate of draft reasoning steps during verification (Acpt.), and execution latency speedup.

### 4.1. Evaluation Results

Table 1 summarizes the autoregressive accuracy of all candidate models and serves as the baseline for evaluating speculative reasoning. The target models, Qwen3-VL-32B and Qwen3-VL-235B, exhibit strong baseline performance, achieving 76.00% and 85.40% on MathVerse, 83.40% and 83.20% on MMBench, and over 75.00% on RealWorldQA. In contrast, draft models are substantially weaker, with Qwen3-VL-2B and Qwen3-VL-4B reaching only 53.94% and 65.40% on MathVerse, respectively. This performance gap highlights the challenge of effective speculative reasoning when draft models differ significantly from the target.

Table 2 demonstrates that our framework preserves accuracy close to the vanilla baseline while delivering substantial acceleration. Using Qwen3-VL-32B as the target and Qwen3-VL-2B as the draft, the supervised variant achieves 74.30% on MathVerse and 80.32% on MMBench, with speedups ranging from 1.8× to 2.2× across benchmarks. The RL-enhanced variant further improves acceptance, yielding speedups of up to 2.38× on MathVerse and 2.48× on MM-Bench and RealWorldQA, while maintaining high accuracy. In contrast, SpecReason underperforms in the VLM setting because its verification relies on discrete, score-based judgments that are brittle under multimodal uncertainty. Small draft models are particularly prone to hallucinations, and the resulting noisy scores can interfere with the target verifier. Moreover, discrete scoring often produces closely clustered or extreme values, making threshold-based accept/reject decisions highly unstable. These factors jointly lead to low acceptance rates and degraded final accuracy.

### 4.2. Ablation Studies

#### 4.2.1. ABLATION ON REWARD FUNCTION DESIGN

Based on Equation 3, we examine how different weighting schemes for draft alignment reward $R_{\text{draft}}$, the outcome reward $R_{\text{outcome}}$, and length penalty reward term $R_{\text{length}}$ affect

the behavior of the draft model. Unless stated otherwise, the baseline setting uses equal weights for all three components, and we vary the relative importance $w_1, w_2, w_3$ of each term to analyze its impact. According to Figure 5, the balanced weighting strategy with $w_1 = 1$, $w_2 = 1$, and $w_3 = 1$ delivers the best overall performance across benchmarks. Specifically, on MathVerse it achieves 75.98% accuracy with a 52.61% acceptance rate and a 2.38× speedup, while on MMBench it attains 82.65% accuracy with a 73.52% acceptance rate and a 1.86× speedup.

Increasing the weight $w_2$ on $R_{\text{draft}}$ while keeping other terms fixed leads to only marginal accuracy changes on MathVerse (77.12%) and a modest increase in acceptance (54.54%), but with reduced speedup (2.31×). On MMBench, acceptance instead drops to 64.15%, suggesting that emphasizing verification alignment alone does not consistently improve verifier agreement in multimodal reasoning.

Adjusting the balance between $R_{\text{outcome}}$ and $R_{\text{length}}$ introduces more pronounced trade-offs. Emphasizing $R_{\text{outcome}}$ increases MathVerse accuracy to 78.80%, but substantially lengthens generation (2906 tokens) and reduces speedup (2.16×). In contrast, strengthening $R_{\text{length}}$ shortens outputs on MathVerse (1668 tokens) but lowers acceptance to 52.26% and causes a sharp acceptance drop on MMBench (38.78%). These results indicate that while outcome-level supervision is beneficial, overly aggressive or imbalanced regularization can negatively affect verifier acceptance.

Employing a stronger draft model (Qwen3-VL-4B) further improves both accuracy and acceptance, resulting in more stable speculative decoding. Across all benchmarks, our approach consistently outperforms existing baselines: LR achieves moderate acceleration but often at the cost of accuracy, while SpecReason suffers from low acceptance when the draft is substantially weaker. In contrast, DREAM-R-NS and DREAM-R simultaneously maintains high accuracy and high acceptance, achieving the most favorable balance between efficiency and correctness.

Scaling to the larger Qwen3-VL-235B target shows that DREAM-R-NS and DREAM-R generalize well across model sizes. When using Qwen3-VL-2B as the draft, DREAM-R-NS achieves speedups ranging from 1.6× to 2.3× while maintaining accuracy close to the vanilla Qwen3-VL-235B baseline, which spans 71.59 to 85.40%. The RL-enhanced variant further improves acceptance by approximately 15%–25%, leading to additional acceleration across all benchmarks. With Qwen3-VL-4B as the draft, accuracy closely matches the vanilla target performance on MMBench and RealWorldQA, while speculative efficiency remains robust, delivering speedups of 1.5× to 1.8×.

Overall, the results demonstrate that our speculative decoding framework consistently achieves the best trade-off

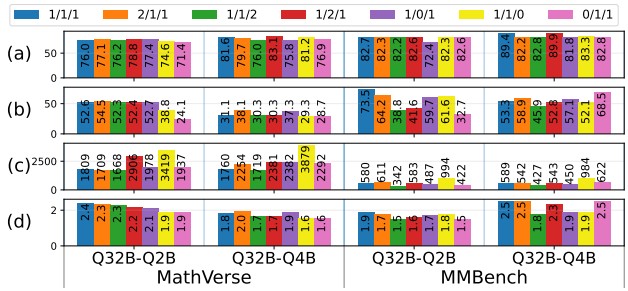

*Figure 5.* Impact of RL reward weighting on (a) output accuracy, (b) reasoning acceptance rate, (c) reasoning token length, and (d) speedup. Values separated by "/" indicate the weights $w_1, w_2, w_3$.

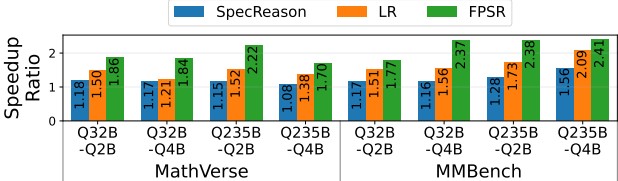

*Figure 6.* Impact of different scheduling methods.

among accuracy retention, acceptance rate, and decoding efficiency. Both the supervised and RL-enhanced variants preserve accuracy close to the vanilla baseline in Table 1 while delivering substantial and stable speedups across four multimodal reasoning benchmarks.

### 4.2.2. ABLATION ON FPSR

We analyze the impact of FPSR by comparing it with two baselines: SpecReason (Pan et al., 2025), LR (Fu et al., 2025), and FPSR. SpecReason does not incorporate explicit scheduling mechanisms and instead directly follows the baseline scheduling strategy illustrated in Figure 4(a). In contrast, LR allows the draft and target models to generate in parallel, overlapping their computation to improve hardware utilization and reduce end-to-end latency. However, verification is deferred until generation finishes, delaying feedback and preventing early rejection of incorrect draft steps. As shown in Figure 6, for the Qwen3-VL-2B to Qwen3-VL-32B setting, speedup increases from 1.18× and

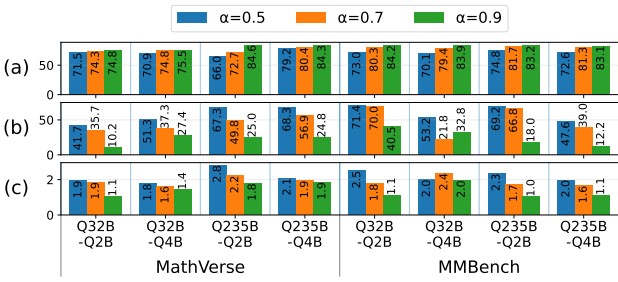

*Figure 7.* Impact of $\alpha$ on (a) accuracy (b) reasoning acceptance rate and (c) speedup.

*Table 2.* Performance evaluations across different models and datasets.

| Target | Draft | Method | MathVerse | | | MMBench | | | RealWorldQA | | | MMMU | | |
|---|---|---|---|---|---|---|---|---|---|---|---|---|---|---|
| | | | Acc.% | Acpt.% | Speedup | Acc.% | Acpt.% | Speedup | Acc.% | Acpt.% | Speedup | Acc.% | Acpt.% | Speedup |
| Q32B | Q2B | Standard SD | 75.99 | – | 1.09× | 79.11 | – | 1.15× | 69.72 | – | 1.12× | 76.15 | – | 1.15× |
| | | SpecReason (Pan et al., 2025) | 44.57 | 14.60 | 1.26× | 62.49 | 24.26 | 1.43× | 39.80 | 21.35 | 1.41× | 46.88 | 16.04 | 1.21× |
| | | LR (Fu et al., 2025) | 72.36 | 35.21 | 1.53× | 74.52 | 71.75 | 1.73× | 72.97 | 66.85 | 2.13× | 75.57 | 41.63 | 1.72× |
| | | DREAM-R-NS | 74.30 | 35.70 | 1.86× | 80.32 | 69.99 | 1.77× | 75.55 | 60.06 | 2.19× | 77.85 | 42.46 | 2.03× |
| | | DREAM-R | 75.98 | 52.61 | 2.38× | 92.65 | 63.52 | 1.86× | 76.10 | 67.98 | 2.48× | 85.79 | 56.44 | 2.19× |
| | Q4B | Standard SD | 71.31 | – | 1.04× | 83.90 | – | 1.11× | 73.48 | – | 1.08× | 77.18 | – | 1.10× |
| | | SpecReason | 40.22 | 20.48 | 0.94× | 73.58 | 32.13 | 1.28× | 69.86 | 48.51 | 1.23× | 53.73 | 32.86 | 1.24× |
| | | LR | 72.08 | 40.28 | 1.55× | 76.41 | 48.92 | 1.58× | 72.12 | 49.29 | 1.66× | 58.96 | 34.35 | 1.66× |
| | | DREAM-R-NS | 74.80 | 37.33 | 1.64× | 79.42 | 49.15 | 2.37× | 81.38 | 47.43 | 1.71× | 66.00 | 34.23 | 1.84× |
| | | DREAM-R | 81.60 | 31.13 | 1.84× | 89.44 | 53.28 | 2.48× | 81.39 | 52.36 | 1.80× | 77.37 | 42.37 | 1.91× |
| | M7B-RL | Standard SD | 73.20 | – | 1.02× | 86.90 | – | 1.05× | 71.10 | – | 1.03× | 74.00 | – | 1.04× |
| | | SpecReason | 48.30 | 18.20 | 0.86× | 45.20 | 15.10 | 0.80× | 51.60 | 23.40 | 0.83× | 50.10 | 27.90 | 0.84× |
| | | LR | 72.10 | 32.50 | 1.40× | 84.30 | 35.60 | 1.32× | 70.50 | 39.70 | 1.38× | 69.20 | 37.30 | 1.36× |
| | | DREAM-R-NS | 76.50 | 34.10 | 1.55× | 88.10 | 60.20 | 1.48× | 73.00 | 41.50 | 1.52× | 71.10 | 39.80 | 1.49× |
| | | DREAM-R | 79.80 | 36.70 | 1.70× | 90.40 | 63.80 | 1.62× | 75.20 | 44.00 | 1.66× | 73.50 | 42.10 | 1.61× |
| | R4B | Standard SD | 72.10 | – | 1.07× | 82.10 | – | 1.09× | 72.10 | – | 1.09× | 68.15 | – | 1.08× |
| | | SpecReason | 46.35 | 21.80 | 1.27× | 71.20 | 29.30 | 1.24× | 62.10 | 38.70 | 1.26× | 54.30 | 40.50 | 1.22× |
| | | LR | 71.25 | 43.90 | 1.67× | 77.35 | 47.80 | 1.78× | 70.95 | 53.80 | 1.82× | 67.10 | 46.90 | 1.80× |
| | | DREAM-R-NS | 74.10 | 45.60 | 1.83× | 80.15 | 52.60 | 2.18× | 72.80 | 50.90 | 1.93× | 69.05 | 49.20 | 1.88× |
| | | DREAM-R | 78.45 | 49.70 | 1.96× | 83.60 | 57.10 | 2.31× | 75.25 | 56.10 | 2.03× | 71.40 | 53.60 | 1.98× |
| Q235B | Q2B | Standard SD | 82.22 | – | 1.11× | 83.20 | – | 1.06× | 75.17 | – | 1.16× | 71.59 | – | 1.07× |
| | | SpecReason | 55.10 | 41.55 | 1.47× | 73.29 | 68.47 | 1.13× | 67.87 | 68.75 | 1.30× | 56.71 | 35.65 | 1.07× |
| | | LR | 66.47 | 48.53 | 1.71× | 80.50 | 66.42 | 1.93× | 74.51 | 52.12 | 1.61× | 71.11 | 36.66 | 1.36× |
| | | DREAM-R-NS | 72.70 | 49.80 | 2.22× | 81.73 | 66.80 | 2.28× | 75.17 | 57.29 | 2.13× | 71.59 | 37.20 | 1.52× |
| | | DREAM-R | 80.00 | 51.82 | 2.28× | 88.45 | 71.45 | 2.31× | 77.89 | 68.05 | 2.26× | 78.33 | 39.21 | 1.60× |
| | Q4B | Standard SD | 81.24 | – | 1.11× | 81.31 | – | 1.09× | 75.36 | – | 1.10× | 69.83 | – | 1.11× |
| | | SpecReason | 47.35 | 29.91 | 1.23× | 69.74 | 39.18 | 1.22× | 55.43 | 36.88 | 1.23× | 59.72 | 51.86 | 1.22× |
| | | LR | 70.70 | 57.65 | 1.68× | 79.10 | 37.34 | 1.45× | 69.88 | 48.22 | 1.48× | 68.20 | 53.70 | 1.48× |
| | | DREAM-R-NS | 80.40 | 56.94 | 1.77× | 81.32 | 39.01 | 1.65× | 68.52 | 44.23 | 1.51× | 71.69 | 53.29 | 1.55× |
| | | DREAM-R | 84.00 | 57.96 | 1.79× | 82.20 | 37.96 | 1.73× | 72.53 | 47.22 | 1.53× | 72.53 | 57.82 | 1.58× |
| | M7B-RL | Standard SD | 81.20 | – | 1.08× | 82.70 | – | 1.04× | 75.10 | – | 1.07× | 71.50 | – | 1.07× |
| | | SpecReason | 64.40 | 36.52 | 0.80× | 17.10 | 14.82 | 0.71× | 11.30 | 24.79 | 1.09× | 48.60 | 23.77 | 0.63× |
| | | LR | 72.10 | 37.91 | 1.46× | 74.70 | 35.46 | 1.46× | 73.30 | 25.24 | 1.40× | 79.80 | 23.03 | 1.45× |
| | | DREAM-R-NS | 78.00 | 36.80 | 1.62× | 79.60 | 35.45 | 1.62× | 76.10 | 25.56 | 1.56× | 81.00 | 23.69 | 1.65× |
| | | DREAM-R | 82.10 | 40.20 | 1.78× | 82.30 | 47.00 | 1.78× | 78.40 | 28.50 | 1.72× | 83.10 | 25.00 | 1.75× |
| | R4B | Standard SD | 80.10 | – | 1.10× | 80.95 | – | 1.08× | 74.20 | – | 1.09× | 68.05 | – | 1.10× |
| | | SpecReason | 49.20 | 27.40 | 1.21× | 67.10 | 37.50 | 1.20× | 57.10 | 35.60 | 1.21× | 60.10 | 49.80 | 1.21× |
| | | LR | 69.85 | 54.20 | 1.65× | 78.25 | 35.80 | 1.43× | 69.05 | 46.90 | 1.47× | 67.30 | 51.90 | 1.46× |
| | | DREAM-R-NS | 79.10 | 53.70 | 1.74× | 79.85 | 37.90 | 1.62× | 67.95 | 42.70 | 1.49× | 70.25 | 50.70 | 1.54× |
| | | DREAM-R | 82.35 | 55.90 | 1.78× | 81.95 | 36.90 | 1.71× | 71.80 | 45.90 | 1.52× | 71.95 | 54.10 | 1.57× |

1.17× with SpecReason (Pan et al., 2025) on MathVerse and MMBench to 1.50× and 1.51× with FPSR, and further to 1.86× and 1.77× under full parallelism. With a stronger draft model (Qwen3-VL-4B), FPSR achieves speedups of 1.84× and 2.37× on the two benchmarks. When the target is Qwen3-VL-235B, the benefit becomes even more significant. For example, with a 2B draft, speedup increases from 1.15× and 1.28× to 1.52× and 1.73× under LR (Fu et al., 2025), and to 2.22× and 2.38× under FPSR.

### 4.2.3. ABLATION ON ACCEPTANCE THRESHOLD FOR CPN

We analyze how the threshold $\alpha$ in CPN (Section 3.2) affects the performance of DREAM-R. By comparing $\phi$ with $\alpha$, this threshold controls the acceptance ratio of drafted reasoning steps. A lower $\alpha$ makes the verifier more permissive, leading to higher acceptance rates and the largest speedups across model pairs, but at the cost of reduced accuracy due to accepting lower-quality reasoning. In contrast, a higher $\alpha$ makes the verifier more conservative, improving accuracy while increasing latency. As shown in Figure 7, setting $\alpha$ to 0.7 achieves the best overall trade-off between accuracy and speedup across different models and datasets. This indicates that the acceptance threshold serves as an effective control knob for tuning system behavior: lower thresholds prioritize efficiency, higher thresholds prioritize accuracy, and intermediate thresholds strike a strong balance.

## 5. Conclusion

We propose *DREAM-R*, a multimodal speculative reasoning framework that accelerates reasoning-heavy decoding while preserving target-model accuracy. Experiments on four multimodal reasoning benchmarks show up to 2.48× speedup with accuracy comparable to vanilla target decoding, outperforming prior speculative reasoning baselines.

## Impact Statement

*DREAM-R* aims to improve the efficiency and scalability of multimodal reasoning systems through speculative reasoning and reinforcement learning based alignment techniques. By reducing inference latency and computational cost, the proposed approach may help make advanced AI systems more accessible and energy-efficient.

At the same time, the underlying foundation models may still inherit limitations such as hallucinations, biased outputs, or incorrect reasoning. As with other large-scale AI systems, the techniques presented in this work could potentially be integrated into high-throughput automated decision-making or content generation pipelines. We therefore encourage responsible deployment and appropriate human oversight when applying such systems in real-world settings.

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

# A. Training Details

## A.1. Setup

*Table 3.* Key hyperparameters used in DREAM-R. Rollout parameters control speculative reasoning behavior, training parameters define draft model optimization, and reward parameters specify the SAPO objective.

| Category | Hyperparameter | Value | Description |
|---|---|---|---|
| Rollout | Acceptance threshold | 0.7 | Confidence threshold for accepting draft steps |
| Rollout | Max prompt length | 2048 | Maximum input context length |
| Rollout | Max response length | 8192 | Maximum generated reasoning length |
| Training | Batch size | 64 | Number of samples per optimization step |
| Training | Learning rate | $1 \times 10^{-6}$ | Step size for parameter updates |
| Training | Epochs | 15 | Number of full passes over training data |
| Training | Precision | BF16 | Numeric precision for training |
| Reward | Outcome reward weight | 1.0 | Weight on final answer correctness |
| Reward | Draft alignment reward weight | 1.0 | Weight on target verification alignment |
| Reward | Length penalty reward weight | 1.0 | Weight on discouraging long reasoning |

Table 3 summarizes the key hyperparameters used in DREAM-R. For decoding, we fix the acceptance threshold to 0.7 and limit the maximum prompt and response lengths to 2048 and 8192 tokens, respectively, ensuring stable speculative verification while bounding the cost of long reasoning traces.

All models are trained with a global batch size of 64 and a learning rate of $1 \times 10^{-6}$ using the Adam optimizer. Training is performed for 15 epochs with BF16 precision, which provides a good balance between numerical stability and computational efficiency.

SAPO employs a composite reward with equal weighting across its components. Specifically, the outcome reward encourages final answer correctness, the draft alignment reward promotes agreement with target verification, and a length penalty discourages excessively long responses through an overlong buffer mechanism. Unless otherwise stated, these hyperparameters are shared across all experiments.

## A.2. Training Cost

Training a single Qwen3-VL-4B draft model with SAPO requires approximately 74 hours on 8×NVIDIA H200 GPUs. The target model is queried via API during rollout and verification. This setup provides a practical balance between training stability and computational cost, enabling scalable reinforcement learning for multimodal speculative reasoning.

# B. Prompts for DREAM-R

We design two complementary prompts in DREAM-R to separate reasoning generation from verification. The *Reasoning Prompt* (Prompt B) guides both the draft and target models to produce structured reasoning, while the *Scoring Prompt* (Prompt B) is used exclusively by the target model to evaluate correctness and alignment.

---

**Reasoning Prompt**

You are a reasoning agent responsible for generating a single coherent reasoning step toward solving the given problem.
**Input**
`{problem}`
`{image},{options}`
**Instructions**

- Produce exactly one reasoning step.

- The step must logically follow from all previous steps.

- Do not generate the final answer unless explicitly required.

---

**Output Format**

```
<reasoning_step>
```

**Scoring Prompt**

You are a verification agent evaluating the correctness of the final reasoning step.
**Input**

- Full problem description

- All previous reasoning steps

- The final candidate step

**Decision Rules**

- Reply `positive` only if the step is factually correct and logically valid.

- Reply `negative` otherwise.

**Output Format**

```
positive | negative
```

