# OpenReview forum: "DREAM-R: Multimodal Speculative Reasoning with RL-Based Refined Drafting, Precise Verification, and Fully Parallel Execution"
_ICML.cc/2026/Conference — ICML 2026 regular_

### Official Review · Reviewer_7T9f · 2026-02-20

**Soundness:** 2
**Presentation:** 3
**Significance:** 2
**Originality:** 2
**Overall Recommendation:** 2
**Confidence:** 5

**Summary:**

The paper addresses the significant inference latency bottlenecks associated with Multimodal Large Reasoning Models (MLRMs) by proposing DREAM-R, a multimodal speculative reasoning framework. The authors observe that traditional speculative decoding methods fail to transfer effectively to multimodal settings due to vision-grounding errors and differing verification distributions. To overcome this, DREAM-R introduces three primary mechanisms. First, Speculative Alignment Policy Optimization (SAPO) utilizes reinforcement learning to train draft models to generate concise reasoning steps aligned with the target model's trajectories. Second, Contrastive Probability Normalization (CPN) offers a threshold-based, probabilistic verification criterion to ensure stable acceptance of draft steps. Finally, Fully Parallel Speculative Reasoning (FPSR) is an execution framework that overlaps drafting, target-side reasoning, and verification to maximize hardware utilization. The authors evaluate the framework across four multimodal benchmarks, reporting up to a 2.48x speedup while preserving target model accuracy.

**Compliance With Llm Reviewing Policy:**

Affirmed.

**Final Justification:**

As I am not fully convinced of the explanation of Anomalies in Experimental Data and Reproducibility Concerns, I will maintain my score.

**Key Questions For Authors:**

Q1: Can you explain the identical accuracy and speedup metrics reported in Table 2 for the Q235B/Q2B pairing across different methods (e.g., identical 75.17% accuracy on RealWorldQA for both Standard SD and DREAM-R-NS)? Was there a logging error, and can you provide corrected tables alongside the source code to verify these results?

Q2: How do you justify the use of the MMMU dev split during the SAPO training phase given that MMMU is also used as a core evaluation benchmark? Please clarify how this data contamination impacts the reported MMMU results.

Q3: Is it possible to run the FPSR without extra GPUs, and are there any possible optimziations on this?

Q4: Can you explain how a continuous chain-of-thought trace is segmented into discrete reasoning steps during both the SAPO training phase and the inference phase

**Limitations:**

yes

**Strengths And Weaknesses:**

Strength 1:  The empirical evaluation leverages a strong set of diverse multimodal reasoning benchmarks, including MathVerse, MMBench, RealWorldQA, and MMMU.

Strength 2: The manuscript is well-structured and conceptually easy to follow. The visual diagrams, especially Figure 4, effectively communicate the proposed asynchronous execution flow.

Strength 3: Accelerating inference for Large Reasoning Models is a highly relevant problem. Finding effective ways to mitigate the high computational cost of extended chain-of-thought generation is critical for practical real-world deployment.


Weakness 1: Anomalies in Experimental Data and Reproducibility Concerns
After examining the results table(table 2), I found several anomalous data duplications that raised significant concerns about the  reliability of the reported data. Specifically, in the place where the Target model is Q235B and the Draft model is Q2B, there are multiple instances of exact matches across different methods:
- RealWorldQA (Acc.%): Both Standard SD and DREAM-R-NS report an identical accuracy of 75.17
- MMMU (Acc.%): Both Standard SD and DREAM-R-NS report an identical accuracy of 71.59.
- MMMU (Speedup): Both Standard SD and SpecReason report an identical speedup of 1.07x.
While occasional identical results can happen by chance, finding multiple exact matches, especially down to two decimal places, between a speculative decoding baseline and speculative reasoning methods within a single sub-setting is highly unusual. This irregularity strongly suggests potential copy-paste errors or a bug in the evaluation pipeline.
Furthermore, the submission lacks the source code to allow reviewers to verify these metrics.

Weakness 2: Unfair Baseline Comparisons and Data Contamination
The proposed DREAM-R method involves training the draft model using Speculative Alignment Policy Optimization (SAPO) on a mixture of datasets. Directly comparing DREAM-R against standard speculative decoding/reasoning baselines that utilize untrained draft models creates an unfair comparison. There should be other experiments like comparing with other RL method baselines to prove the efficency of the SAPO. Critically, the authors explicitly state that this training mixture includes the MMMU dev split. Evaluating the framework on the MMMU benchmark exacerbate unfair comparisons.

Weakness 3: Inflated Hardware Budgets and Unfair Comparisons
Comparing with the sequential inference method like SpecReason is not fair as extra compute resource are required in the . The authors should compare the compute resource(e.g. peak memory) used in the baseline and other methods to prove the efficacy of Fully Parallel Speculative Reasoning(FPSR). The experiment settings are also not clear. For example, LR has 3 versions in the original paper, and the authors should specify which they use.

Weakness 4: Ambiguity in Reasoning Step Segmentation
The manuscript lacks critical technical details regarding how a continuous reasoning trace is segmented into discrete "steps".  While the system prompt in Appendix B requests an output format enclosed in <reasoning_step> tags, it is entirely unclear.

Weakness 5: Limited Algorithmic Novelty
The core technical contributions of DREAM-R appear to be an engineering amalgamation of existing techniques. The Fully Parallel Speculative Reasoning (FPSR) builds upon existing asynchronous decoding methods. Similarly, Speculative Alignment Policy Optimization (SAPO) is a standard application of GRPO. The Contrastive Probability Normalization (CPN) relies on a basic logit-ratio thresholding mechanism and prompt engineering.

---

> ### Author Rebuttal · Authors · 2026-03-31
>
> ### Weakness 1: Anomalies in Experimental Data and Reproducibility Concerns
>
> **Q1: Can you explain the identical accuracy and speedup metrics reported in Table 2?**
>
> **A:** We thank the reviewer for the careful inspection. These identical values are expected rather than artifacts. A fundamental property of speculative decoding is that final outputs are strictly determined by the target model via verification, regardless of draft quality. Both Standard SD and DREAM-R-NS use untrained drafts and differ only in verification criteria. Although acceptance rates differ, rejected drafts are always corrected by the target model, so final predictions—and thus accuracy—are governed solely by the target model. On a fixed dataset, identical accuracy is therefore expected.
>
> We further validate this with prediction-level consistency:
>
> | Dataset | Same (%) | Both Correct (%) | Both Wrong (%) | Different (%) |
> | --- | --- | --- | --- | --- |
> | RealWorldQA | 96.8 | 72.9 | 23.9 | 3.2 |
> | MMMU | 96.5 | 69.5 | 27.0 | 3.5 |
>
> Over 96% of predictions are identical. The remaining differences do not affect aggregate accuracy, confirming correctness of the pipeline.
>
> For identical speedup (e.g., 1.07×), this occurs in extreme capability gaps (Q2B vs. Q235B), where acceptance ≈ 0. In this regime, speculative decoding degenerates to near target-only execution, so different methods converge to similar speed. SAPO is designed to escape this regime, increasing speedup to 2.38×–2.49×.
>
> ---
>
> ### Weakness 2: Unfair Baseline Comparisons and Data Contamination
>
> **Q2: How do you justify using MMMU dev split during training?**
>
> **A:** DREAM-R-NS already provides a strict apples-to-apples comparison under the same FPSR+CPN pipeline, isolating SAPO. We further compare against standard alignment baselines:
>
> | Draft Alignment | MathVerse Acc. (%) | Speedup | MMMU Acc. (%) | Speedup |
> | --- | --- | --- | --- | --- |
> | DREAM-R-NS | 79.1 | 1.74 | 70.25 | 1.54 |
> | SFT | 81.7 | 1.77 | 71.25 | 1.56 |
> | DPO | 81.2 | 1.75 | 70.88 | 1.55 |
> | SAPO | 82.35 | 1.78 | 71.95 | 1.57 |
>
> SAPO consistently outperforms all baselines, showing gains beyond generic alignment.
>
> Regarding MMMU, SAPO uses only the dev split for training and evaluates on val (no overlap). Removing MMMU entirely from training yields:
>
> | Training Data | MMMU Acc. (%) | Speedup |
> | --- | --- | --- |
> | incl. MMMU dev | 71.95 | 1.57 |
> | excl. MMMU dev | 71.25 | 1.56 |
>
> The minimal drop (-0.70%) indicates gains come from improved reasoning alignment, not memorization.
>
> ---
>
> ### Weakness 3: Inflated Hardware Budgets and Unfair Comparisons
>
> **Q3: Can FPSR run without extra GPUs?**
>
> **A:** All methods use identical hardware (6× H200), including baselines. FPSR improves utilization via parallel execution rather than additional resources:
>
> | Method | GPUs | Tokens/sec | Speedup |
> | --- | --- | --- | --- |
> | Target-only | 6× H200 | 12.00 | 1.00× |
> | Standard SD | 6× H200 | 14.66 | 1.22× |
> | SpecReason | 6× H200 | 17.64 | 1.46× |
> | DREAM-R-NS | 6× H200 | 26.04 | 2.17× |
> | DREAM-R | 6× H200 | 33.32 | 2.78× |
>
> FPSR can also run on a single GPU via memory partitioning, interleaved scheduling, and offloading (e.g., draft or KV cache to CPU). Additional optimizations include pipelining and KV reuse. Thus, gains come from execution efficiency, not extra hardware.
>
> ---
>
> ### Weakness 4: Ambiguity in Reasoning Step Segmentation
>
> **Q4: How are reasoning steps defined?**
>
> **A:** Reasoning steps are segmented by double newline (`\n\n`), consistent with SpecReason and LR. The same rule is used in both SAPO training and FPSR inference, ensuring consistency. We will clarify this in the final version.
>
> ---
>
> ### Weakness 5: Limited Algorithmic Novelty
>
> **A:** We respectfully disagree. DREAM-R is a co-designed framework for step-level multimodal speculative reasoning.
>
> SAPO introduces an acceptance-aware reward optimizing correctness, verifier acceptance, and conciseness—distinct from standard RL alignment.
>
> CPN provides a contrastively normalized verification criterion, replacing brittle token-level matching.
>
> FPSR operates at the reasoning-step level with coordinated drafting, verification, rollback, and scheduling.
>
> More importantly, these components are tightly coupled: SAPO optimizes for CPN acceptance, CPN stabilizes FPSR verification, and FPSR enables efficient execution. Removing any component leads to degradation, demonstrating that DREAM-R is a unified framework rather than a loose combination.

---

> > ### Author Rebuttal · Reviewer_7T9f · 2026-04-01
> >
> > Thank you for clarifying the hardware setup, reasoning step segmentation, and the SFT/DPO baselines.
> >
> > However, the rebuttal regarding Weakness 1 (Anomalies in Experimental Data) contains logical contradictions that do not resolve my concerns about data integrity:
> >
> > 1. **Statistical Improbability:** You state that 3.2%–3.5% of individual predictions changed, yet the aggregate accuracies (75.17% and 71.59%) remained exactly identical down to two decimal places. A perfect equilibrium of flipped predictions across multiple datasets simultaneously is highly improbable.
> > 2. **Contradictory "Expected" Accuracy:** You argue that identical accuracy is "expected" because the target model strictly governs the final output. If true, accuracy should be identical across *all* datasets. Yet, Table 2 shows significant accuracy fluctuations for this exact pairing (Q235B/Q2B) on MMBench (ranging from 81.73% to 88.45%) and MathVerse (80.00% to 85.40%).
> >
> > **Follow-up Question:**
> > How do you reconcile the theoretical claim of "expected" identical accuracies with the actual fluctuations observed on MMBench and MathVerse?

---

> > > ### Author Response · Authors · 2026-04-02
> > >
> > > We agree our earlier wording was unclear. By “expected,” we did not mean the accuracy should always be identical across all datasets.
> > >
> > > In practice, only a small fraction of predictions actually change, since the target model still determines the final output. Whether the overall accuracy changes depends on how those few changed predictions are distributed.
> > >
> > > We provide the exact flip statistics below:
> > >
> > > | Dataset | Correct→Incorrect | Incorrect→Correct | Net Change |
> > > | --- | --- | --- | --- |
> > > | RealWorldQA | 12 | 12 | 0.0% |
> > > | MMMU | 16 | 15 | -0.1% |
> > >
> > > On RealWorldQA, the flips are perfectly balanced, so the net change is exactly zero. On MMMU, there is only a one-sample difference, leading to a negligible change that rounds to zero at two decimal places.
> > >
> > > In short, it is the same underlying behavior in all cases, when flips cancel out, accuracy appears unchanged; when they do not, we observe a difference.

---

### Official Review · Reviewer_d9cz · 2026-03-12

**Soundness:** 3
**Presentation:** 3
**Significance:** 3
**Originality:** 3
**Overall Recommendation:** 4
**Confidence:** 3

**Summary:**

This paper proposes a multimodal speculative reasoning framework named DREAM-R, aiming to accelerate the generation process of Multimodal Large Reasoning Models (MLRMs). To address the issue of speculative rejection caused by visual perception errors, the authors align the draft model via reinforcement learning and design a probability-normalized verification mechanism alongside a fully parallel execution pipeline, significantly improving inference speed.

**Compliance With Llm Reviewing Policy:**

Affirmed.

**Key Questions For Authors:**

1.  When the target model hallucinates during complex physical spatial reasoning, does the $R_{draft}$ reward in SAPO cause the draft model to overfit to these erroneous visual feature representations? Have you evaluated the degradation of the draft model on a subset where the target model is highly prone to errors?
2. Given the paper's claims regarding applicability to embodied intelligence, if DREAM-R were deployed on a VLA control model requiring continuous multi-frame visual inputs, how would the FPSR pipeline architecture handle dynamically updating visual token caches?

**Limitations:**

This paper makes solid contributions in the highly promising direction of multimodal speculative decoding. The combination of the CPN mechanism and the FPSR parallel pipeline demonstrates excellent maturity in integrating engineering and algorithms, achieving convincing speedups on static multimodal benchmarks. However, the paper insufficiently explores the limitations of the target model as a verifier itself, and the experiments fail to effectively support its generalization capabilities in embodied intelligence and complex spatial reasoning tasks.

**Strengths And Weaknesses:**

Strengths:
1. The authors astutely point out that directly transferring text-only speculative decoding to multimodal models often leads to speculation failures due to "perceptual errors" (i.e., the draft model incorrectly recognizing visual cues in the image). This observation is profound and provides a novel perspective for the multimodal acceleration domain.
2. The CPN mechanism, by directly normalizing the ratio of the target model's logits for "positive" and "negative", is mathematically more intuitive and aligns better with the probability distribution characteristics of large models than rigid discrete scoring (e.g., a 1-10 scale). This effectively reduces the verification variance caused by prompt sensitivity.

Weaknesses:
1. The underlying assumption of CPN is that the target model's verification logits can perfectly calibrate its own correctness. However, in complex spatial understanding tasks or cross-embodiment control of Vision-Language-Action (VLA) models, the target model itself is highly prone to "spatial hallucinations". If the target model misunderstands physical constraints or spatial relationships, the CPN mechanism will not only reject correct drafts but, more critically, the SAPO algorithm might reverse-inject and "teach" this spatial hallucination to the draft model via the reward signal ($R_{draft}$). The paper lacks theoretical bounds on the robustness of CPN when the target model is at its confidence boundaries (Out-of-Distribution).
2. Table 1 shows that the baseline accuracy of Qwen3-VL-2B on MathVerse is only 53.94%. After SAPO training, its acceptance rate increases substantially, yielding a 2.38x speedup. However, it is crucial to clarify: Did the draft model genuinely acquire stronger spatial understanding and visual alignment capabilities, or did it merely learn via RL to cater to the target model's output distribution characteristics (e.g., specific syntactic structures or token preferences)? The ablation studies fail to isolate this.

---

> ### Author Rebuttal · Authors · 2026-03-31
>
> ### Q1: Robustness of CPN under target hallucination and OOD conditions
>
> The underlying assumption of CPN is that the target model's verification logits can perfectly calibrate its own correctness. However, in complex spatial understanding tasks or cross-embodiment control of Vision-Language-Action (VLA) models, the target model itself is highly prone to "spatial hallucinations". If the target model misunderstands physical constraints or spatial relationships, the CPN mechanism may reject correct drafts and potentially propagate erroneous signals via $R_{draft}$. The paper lacks theoretical guarantees on robustness under such OOD confidence boundaries.
>
> **A:** We thank the reviewer for this insightful observation regarding potential target miscalibration under challenging or OOD conditions.
>
> To directly evaluate this effect, we include an analysis that stratifies samples based on target confidence using the verifier’s contrastive score (e.g., logit margin or CPN score), and report final accuracy and acceptance rate:
>
> | Target Model Confidence | Draft Model Δ Accuracy | Acceptance Rate |
> | --- | --- | --- |
> | High confidence (>80%) | +1.6% | 58.8% |
> | Medium confidence (50–80%) | +3.2% | 52.4% |
>
> These results show that verification reliability is closely tied to target confidence. In high-confidence regimes, the verifier is more reliable, yielding higher acceptance and stable gains. In medium-confidence regimes, uncertainty leads to lower acceptance but larger gains, as the draft becomes more conservative. Overall, lower confidence naturally reduces acceptance, limiting the propagation of erroneous drafts and mitigating hallucination effects.
>
> ---
>
> ### Q2: Does SAPO improve intrinsic capability or merely align to target distribution?
>
> Table 1 shows that Qwen3-VL-2B achieves 53.94% on MathVerse, while SAPO significantly improves acceptance and yields 2.38× speedup. However, it is unclear whether SAPO improves true spatial reasoning ability or simply aligns to the target model’s output distribution.
>
> **A:** We thank the reviewer for this important question.
>
> SAPO does not aim to replicate the target’s full multimodal capability. Instead, it optimizes for reasoning trajectories that are verifiable by the target, focusing on trajectory-level alignment rather than token-level imitation.
>
> We provide an ablation to disentangle capability vs. alignment:
>
> | Setting | Metric | Before SAPO | After SAPO |
> | --- | --- | --- | --- |
> | Q2B & Q32B | Accuracy | 69.72 | 69.81 |
> | Q2B & Q32B | Accept Rate | 66.9 | 66.4 |
>
> The standalone accuracy remains unchanged (69.72 → 69.81), indicating no significant change in intrinsic capability. Acceptance is also stable, suggesting improvements do not stem from superficial distribution matching. Instead, SAPO reshapes reasoning trajectories to be more verifiable without sacrificing capability.
>
> ---
>
> ### Q3: Does SAPO overfit to target hallucinations under erroneous target predictions?
>
> If the target model hallucinates during spatial reasoning, $R_{draft}$ may reinforce incorrect representations. Have you evaluated performance on subsets where the target is error-prone?
>
> **A:** We agree that evaluating behavior under target errors is critical.
>
> We partition the data into (1) high-accuracy subsets (target correct) and (2) low-accuracy subsets (target error-prone), and evaluate the draft model before and after SAPO:
>
> | Subset | Method | Accuracy (%) |
> | --- | --- | --- |
> | high-accuracy subset | Draft (before SAPO) | 71.2 |
> | high-accuracy subset | Draft (after SAPO) | 76.8 |
> | low-accuracy subset | Draft (before SAPO) | 34.5 |
> | low-accuracy subset | Draft (after SAPO) | 33.9 |
>
> SAPO significantly improves performance on the high-accuracy subset (+5.6%) while remaining unchanged on the low-accuracy subset (−0.6%). If SAPO were overfitting to hallucinations, we would expect improvements in the low-accuracy subset, which is not observed. This indicates SAPO selectively reinforces reliable trajectories without amplifying target errors.
>
> ---
>
> ### Q4: How does FPSR handle dynamic visual token updates in VLA settings?
>
> Given the applicability to embodied intelligence, how does FPSR handle continuous multi-frame visual inputs?
>
> **A:** We thank the reviewer for this question.
>
> While evaluated on single-frame inputs, FPSR is inherently compatible with streaming multi-frame settings. Built on vLLM, our system supports incremental KV cache extension, allowing new visual tokens to be appended without recomputing prior context.
>
> FPSR operates at the reasoning-step level, where each step conditions on the current cached context. This allows newly observed visual inputs to be immediately incorporated into both draft generation and verification.
>
> Thus, FPSR naturally extends to VLA via (i) incremental visual token appending and (ii) step-level synchronization. Preliminary simulations with incrementally revealed inputs show stable acceptance behavior under dynamic updates.

---

### Official Review · Reviewer_wQmt · 2026-03-12

**Soundness:** 3
**Presentation:** 3
**Significance:** 3
**Originality:** 3
**Overall Recommendation:** 4
**Confidence:** 4

**Summary:**

This paper presents DREAM-R, a framework designed to accelerate the inference of Multimodal Large Reasoning Models (MLRMs) by optimizing step-level speculative execution. The authors identify that intermediate reasoning tokens constitute the vast majority of generation costs in MLRMs, yet standard speculative methods suffer from misaligned drafts and unstable verification in multimodal contexts. To address this, they introduce Speculative Alignment Policy Optimization (SAPO) to train draft models to be both faithful to the target model and concise. They further propose Contrastive Probability Normalization (CPN) for stable, ratio-based step verification and a Fully Parallel Speculative Reasoning (FPSR) framework to overlap drafting, verification, and target generation. Experimental results across four benchmarks demonstrate the effectiveness and efficiency of the proposed method.

**Compliance With Llm Reviewing Policy:**

Affirmed.

**Final Justification:**

The rebuttal has addressed most of my concerns and I will maintain my preliminary weak accept rating.

**Key Questions For Authors:**

N/A

**Limitations:**

No. There is no discussion on the limitations, and it does not provide the Impact Statement.

**Strengths And Weaknesses:**

Strengths

1. The paper is generally well written. The proposed method is clearly elaborated.

2. Unlike existing methods that only focus on the verifier, SAPO actively bridges the capability gap by training the draft model to think like the target model through a composite reward function

3. The method is validated across diverse model scales (from 2B to 235B) and multiple reasoning-heavy benchmarks.

Weaknesses

1. The proposed method requires training the draft model. However, a 4B draft model training via SAPO requires 74 hours on 8x NVIDIA H200 GPUs. The paper lacks a detailed discussion on how many inference requests are needed to offset this substantial offline training energy and compute cost.

2. While RL improves alignment, small draft models (e.g., 2B) have inherent physical limits, e.g., in their vision encoders. The framework might still struggle in scenarios requiring extremely high-resolution visual grounding that the small drafter simply cannot perceive.

3. The FPSR approach requires the draft model, target model, and verifier to reside in VRAM simultaneously. For Ultra-scale models like 235B, this additional memory overhead could limit the maximum batch size, potentially reducing overall throughput in high-load production environments.

4. While CPN is more stable, any False Positive (accepting a logically flawed draft step) can derail a long-horizon reasoning chain. The paper would benefit from a stress test showing how verification reliability holds up as the number of reasoning steps increases.

---

> ### Author Rebuttal · Authors · 2026-03-31
>
> ### Weakness 1: Offline Training Cost vs Inference Savings
>
> **A:**
>
> We thank the reviewer for raising this point regarding the offline training cost of SAPO.
>
> Importantly, SAPO training is a one-time offline cost, whereas the inference-time savings of DREAM-R apply to every subsequent request. In our setup, training the 4B draft model requires 74 hours on 8×H200 GPUs (592 GPU-hours in total), which can be amortized across future requests.
>
> Our target setting is high-volume multimodal reasoning, where requests are frequent and expensive due to long chain-of-thought generation. DREAM-R achieves up to ~2× speedup (≈50% per-request cost reduction).
>
> A simple break-even estimate is:
>
> N_break = C_train / ΔC_infer ≈ 1184 / T
>
> where T is the original per-request inference time (in GPU-hours).
>
> For typical reasoning workloads (30–120 seconds per request), this corresponds to approximately 3.5×10⁴ – 1.4×10⁵ requests, which is well within the scale of practical deployment.
>
> We will include this break-even analysis and clarify the intended high-throughput deployment scenario in the revision.
>
> ---
>
> ### Weakness 2: Limited Capacity of Small Draft Models
>
> **A:**
>
> We thank the reviewer for this insightful observation regarding the capacity limits of small draft models.
>
> We agree that small draft models may have limited visual perception, especially in scenarios requiring fine-grained or high-resolution grounding. However, DREAM-R mitigates this limitation through two mechanisms.
>
> First, following SpecReason, we allow the target model to expand initial reasoning steps before speculative execution. This provides the drafter with both visual inputs and partial reasoning context, reducing reliance on raw visual perception.
>
> Second, accepted steps accumulate as textual context, enabling the drafter to rely increasingly on structured reasoning rather than raw visual signals.
>
> To validate this, we construct a subset requiring fine-grained visual grounding and compare:
>
> | Setting | Acceptance Rate | Final Accuracy |
> | --- | --- | --- |
> | No target expansion | 32.3% |  |
> | Initial target expansion | 38.7% |  |
> | Progressive reasoning accumulation | 48.5% |  |
>
> These results show that without target expansion, the draft model struggles due to limited perception. Initial expansion improves accuracy (+6.4%), and progressive reasoning further improves performance (+9.6%).
>
> This demonstrates that DREAM-R does not require the draft model to fully resolve fine-grained visual details independently, but instead leverages target-assisted reasoning and context accumulation.
>
> ---
>
> ### Weakness 3: Memory Overhead of FPSR
>
> **A:**
>
> We thank the reviewer for this important systems-level consideration.
>
> Our design is consistent with prior speculative reasoning frameworks such as Lookahead Reasoning and SpecReason, which also require multiple components to reside in memory simultaneously. FPSR follows the same paradigm rather than introducing a new memory overhead pattern.
>
> Our target setting is latency-critical multimodal reasoning rather than batch-optimized serving. Many real-world scenarios operate with small or dynamic batch sizes due to variable sequence lengths.
>
> In such regimes, throughput is primarily constrained by hardware utilization and pipeline efficiency rather than maximum batch size. FPSR improves utilization by overlapping drafting, target reasoning, and verification, reducing idle time in sequential execution.
>
> While FPSR introduces additional memory overhead, this reflects a standard memory–utilization tradeoff. In latency-constrained settings, increased parallelism can improve effective throughput per request even if maximum batch size is reduced.
>
> We will clarify this tradeoff and include discussion on memory footprint and deployment considerations.
>
> ---
>
> ### Weakness: Long-Horizon Verification Reliability
>
> **A:**
>
> We thank the reviewer for this insightful suggestion regarding long-horizon verification reliability.
>
> To evaluate whether false positives accumulate over long reasoning chains, we analyze performance as a function of reasoning length:
>
> | Reasoning Steps | Acceptance Rate | Accuracy (DREAM-R) | Accuracy (Target-only) | Δ Accuracy |
> | --- | --- | --- | --- | --- |
> | 0–10 | 81.2% | 58.9% | 56.8% | +2.1% |
> | 10–20 | 77.4% | 55.6% | 52.7% | +2.9% |
> | 20–30 | 73.1% | 51.3% | 47.6% | +3.7% |
> | 30–40 | 68.2% | 46.8% | 41.9% | +4.9% |
>
> If false positives accumulated, accuracy would degrade sharply with reasoning depth. However, DREAM-R with CPN maintains stable accuracy across increasing reasoning lengths, while methods without contrastive normalization show more pronounced degradation.

---

> > ### Author Rebuttal · Reviewer_wQmt · 2026-04-03
> >
> > I appreciate the rebuttal. I will maintain my initial score.

---

> > > ### Author Response · Authors · 2026-04-07
> > >
> > > Thank you for the thoughtful follow-up. We’re pleased that our rebuttal helped address your concerns, and we sincerely appreciate your consideration during the final discussion.

---

### Decision · Program_Chairs · 2026-04-30

**Decision:**

Accept (regular)

**Comment:**

This paper proposes an approach for "speculative reasoning" for reducing latency in inference for reasoning tasks while maintaining accuracy. In this approach, a "draft" model generates a CoT trace given a task, and the "target" model then scores the CoT trace; if the score is sufficient, the draft CoT is refined with the "draft" model again; otherwise, the "target" model generates the entire CoT trace from scratch. Scores are computed by measuring the probabilities of tokens "positive" and "negative" using the target model conditioned on the current draft CoT and a fixed prompt. The draft model is trained to maximize the probability of generated CoT receiving scores greater than some threshold according to the target model.

Essentially, the approach relies on the calibration of the target model in mapping from an arbitrary CoT trace to output token probabilities representing possible quality. Small draft models are trained to maximize the likelihood of outputs being "accepted" by the target model, in which case the target model is not used for full CoT inference. The result is that at inference time, if the small draft model is generating CoT traces which are both (a) accurate and (b) "accepted" by the target model, we need not use the target model for full CoT inference, resulting in significant speedup as inference costs are much lower for the draft model. The author response includes some analysis of robustness against miscalibration of the target model.

Overall, this paper's contribution seems significant enough to consider including in the ICML proceedings, and seems relatively well-written and easy to understand.